# Separation Iron(III)-Manganese(II) via Supported Liquid Membrane Technology in the Treatment of Spent Alkaline Batteries

**DOI:** 10.3390/membranes11120991

**Published:** 2021-12-19

**Authors:** Francisco J. Alguacil, Félix A. Lopez

**Affiliations:** Centro Nacional de Investigaciones Metalurgicas (CENIM), Consejo Superior de Investigaciones Científicas (CSIC), Avda. Gregorio del Amo 8, 28040 Madrid, Spain; fjalgua@cenim.csic.es

**Keywords:** membrane transport, Cyanex 923, iron(III), manganese(II), separation

## Abstract

In this paper, the transport of iron(III) from iron(III)-manganese(II)-hydrochloric acid mixed solutions, coming from the treatment of spent alkaline batteries through a flat-sheet supported liquid membrane, is investigated (the carrier phase being of Cyanex 923 (commercially available phosphine oxide extractant) dissolved in Solvesso 100 (commercially available diluent)). Iron(III) transport is studied as a function of hydrodynamic conditions, the concentration of manganese and HCl in the feed phase, and the carrier concentration in the membrane phase. A transport model is derived that describes the transport mechanism, consisting of diffusion through a feed aqueous diffusion layer, a fast interfacial chemical reaction, and diffusion of the iron(III) species-Cyanex 923 complex across the membrane phase. The membrane diffusional resistance (Δ_m_) and feed diffusional resistance (Δ_f_) are calculated from the model, and their values are 145 s/cm and 361 s/cm, respectively. It is apparent that the transport of iron(III) is mainly controlled by diffusion through the aqueous feed boundary layer, this being the thickness of this layer calculated as 2.9 × 10^−3^ cm. Since manganese(II) is not transported through the membrane phase, the present system allows the purification of these manganese-bearing solutions.

## 1. Introduction

Recycling is an issue of increasing necessity in the 21st century which is caused, among other factors, by the decrease and depletion of certain raw materials, and more stringent environmental pressures about the discharge of contaminants.

One example of materials to be recycled are alkaline batteries, which also contain valuable metals that can be recovered and reuse (allowing one to gain a further profit from the recycling process). CENIM-CSIC developed investigations about the use of ammonium carbonate processing in the recycle of zinc-bearing materials such as electric arc furnace (EAF) flue dusts and Waelz oxides [1,2,3]. Since zinc is an important component of alkaline batteries, this type of processing was applied on the recycle of these spent alkaline batteries [4,5]. In the process, the mass yielded after breaking of the battery case is subjected to a leaching process in ammoniacal ammonium carbonate solutions. Zinc is recovered from the leachate, whereas the solid residue contains iron and manganese (VII) compounds. This residue is treated via leaching in HCl medium, yielding a solution containing Fe(III) and Mn(II), since in the operational conditions, manganese(VII) oxidizes chloride ions and, thus, it is reduced to Mn(II). From the solution, Fe(III) and Mn(II) can be separated using liquid-liquid extraction technology. However, the use of liquid membranes, specifically supported liquid membranes technologies, can be also of interest due to the economical and operational advantages that these technologies possess.

In fact, supported liquid membrane technologies have been proposed for the treatment of strategic and toxic metals. Some of their uses have recently been reviewed [6,7] and specifically published in the case of heavy metals [8,9,10], germanium [11], rare earth metals [12,13], scandium [14], europium [15], and tungsten [16].

Before scaling up the flat-sheet supported liquid membrane to a more dynamic membrane methodology, such as hollow fiber modules, a model of the liquid membrane system is convenient in order to design this more efficient process.

The present investigation presents a kinetic modelling of the active transport of Fe(III), from Fe(III)-Mn(II)-HCl solutions, using Cyanex 923-Solvesso 100 solutions immobilized on a microporous hydrophobic support. These aqueous solutions originated in the processing of spent alkaline batteries. Diffusional resistances due to the membrane and the feed phase are calculated from the model, and the influence of hydrodynamic and chemical conditions are established in order to yield an efficient flat-sheet liquid membrane system.

## 2. Materials and Methods

### 2.1. Reagents and Solutions

The extractant used in the investigation was Cyanex 923 (Solvay, Brussels, Belgium, commercially available phosphine oxide), which was dissolved in Solvesso 100 (Exxon Chem Iberia, Barcelona, Spain, aromatic diluent) in order to obtain an adequate range of extractant concentrations to the transport experiments. Also, the composition of the aqueous feed solutions was scaled down to the metal concentrations range convenient to this type of membrane experimentation. All of the chemicals used in the experimentation were of AR grade.

The solid support used in the present work was Millipore Durapore GVHP4700 (Celanese Plastics, Irving, TX, USA) (microporous polyvinylidene fluoride film) of 75% porosity, 1.67 tortuosity, 12.5 × 10^−3^ cm thickness, and 0.22 μm effective pore size.

### 2.2. Methods

#### 2.2.1. Liquid-Liquid Extraction Experiments

Iron (III) liquid-liquid extraction experiments were performed by mixing, in thermostatted separatory funnels, equal volumes of organic solutions of Cyanex 923 (from 1.9 × 10^−2^ to 0.13 M) in Solvesso 100 with 0.01 g/L iron(III) in 4 M HCl solutions for 10 min at 20 °C. The careful optimization of experimental setups indicates that equilibrium reaches within 5 min of contact of both the phases. After the quick phase disengagement (less than 30 s), iron(III) was analyzed in the aqueous solutions by atomic absorption spectrometry, reproducibility within ±3%, and the metal concentration in the equilibrated organic solutions was estimated by the corresponding mass balance.

#### 2.2.2. SLM Experiments

Transport experiments were carried out in a two compartments cell which consisted of a feed solution half-cell (200 cm^3^) separated from the receiving solution half-cell (200 cm^3^) by the solid support with an effective membrane area of 11.3 cm^2^. The feed and the receiving solutions were mechanically stirred at 20 °C to avoid concentration polarization conditions at the support interfaces and in the bulk of both solutions.

The supported liquid membrane was prepared by impregnation of the solid support with the corresponding carrier solution by immersion for 24 h. Then, it was left to drip for 20 s before being placed in the cell.

Metals transport were determined by monitoring Fe(III) and Mn(II) concentrations in the feed and receiving phases as a function of time and by atomic absorption spectrometry. Metal concentrations in the solutions were found to be reproducible within ±3%. The iron overall mass transfer coefficient (K_Fe_) was computed using the next equation:(1)$$\ln\frac{{\left[\mathrm{Fe}\right]}_{f,t}}{{\left[\mathrm{Fe}\right]}_{f,0}}=-\frac{AK_{Fe}}{V}t$$
where A was the support area, V is the volume of the feed solution, [Fe]_f,t_ and [Fe]_f,0_ were the iron concentrations in the feed solution at an elapsed time and time zero, respectively, and t was the elapsed time.

The percentage of iron transported to the receiving solution was calculated by:(2)$$\%T=\frac{{\left[\mathrm{Fe}\right]}_{r,t}}{{\left[\mathrm{Fe}\right]}_{f,0}-{\left[\mathrm{Fe}\right]}_{f,t}}100$$
where [Fe]_r,t_ was the iron concentration in the receiving solution at an elapsed time. Manganese(II) transport was evaluated as above, and under the present experimental conditions, basically manganese(II) was not transported (K_Mn_ < 1 × 10^−6^ cm/s) across the membrane.

## 3. Results

### 3.1. Iron(III) Liquid-Liquid Extraction Equilibrium

The extraction of iron(III) by the phosphine oxide was based on a solvation reaction [17], represented by the next equilibrium [18,19]:(3)$$nL_{org}+FeCl_{4_{aq}}^{-}+H_{aq}^{+}\Leftrightarrow HFeCl_{4}\cdot nL_{org}$$
where L represented the extractant molecule, n was an stoichiometric factor (the number of extractant molecules involved in the extraction process), and the subscripts _aq_ and _org_ referred to the equilibrated aqueous and organic phases, respectively.

The extraction equilibrium can be described by the next equation:(4)$$K_{ext}=\frac{{\left[HFeCl_{4}L_{n}\right]}_{org}}{{\left[FeCl_{4}^{-}\right]}_{aq}{\left[H^{+}\right]}_{aq}{\left[L\right]}_{org}^{n}}$$

In order to validate Equation (3) and calculate both the extraction constant and the stoichiometric factor n values, the experimental data were treated by a tailored computer program which minimized the U function, defined as:(5)$$U=\Sigma {\left(\log D_{cal}-\log D_{\exp}\right)}^{2}$$

D_exp_ and D_cal_ being the experimental distribution coefficients and the corresponding values calculated by the program. The experimental distribution coefficient D_exp_ was calculated as:(6)$$D_{\exp}=\frac{{\left[Fe\right]}_{org}}{{\left[Fe\right]}_{aq}}$$

In the above equation, [Fe]_org_ and [Fe]_aq_ are the iron concentrations in the equilibrated organic and aqueous phases, respectively.

The results derived from the experimental data and treated by the program indicated that the extraction of iron (III) by Cyanex 923 corresponded to the reaction showed in Equation (3), with log K_HCl_ 4.86 and U 0.093. The stoichiometric factor n was calculated as 4.

### 3.2. Iron(III) Transport through the Supported Liquid Membrane

Figure 1 showed a probable transport scheme for Fe(III) with Cyanex 923 (represented as L) dissolved in Solvesso 100 through a supported liquid membrane. The driving force for iron(III) transport was the difference in acidity between the feed and receiving phases, and thus iron(III) transport was coupled to the acid co-transport from the feed to the receiving phase.

#### 3.2.1. Influence of the Stirring Speed Applied on the Feed Phase on Iron(III) Transport

In order to yield effective iron(III) transport through the supported liquid membrane, it was of importance to establish the influence of the hydrodynamic conditions, firstly, and the influence of the stirring speed, applied to the feed phase on the overall mass transfer coefficient, was investigated. The transport of iron(III) across the supported liquid membrane was dominated by diffusional resistances which can be of two types: (i) one associated to the feed phase boundary layer, and (ii) other associated to the membrane support. It is relatively usual that the magnitude of the first competed with the value of the support resistance [20].

In the present work, stirring of the feed solution was carried out from 500 to 1500 min^−1^ (Table 1).

It can be seen that the overall mass transfer value increased with the increase of the stirring speed up to 1250 min^−1^ and then remained constant. Thus, at 1250 min^−1^, the thickness of the aqueous feed boundary layer reached a minimum, and iron(III) transport maximized. Also, from the results showed in this Table 1, it can be observed that iron(III) recovery in the receiving phase was practically quantitative; the transport of Mn(II) can be considered as negligible (see Section 2.2.2). Thus, in the first instance, the present system (carrier phase and support) can be used to separate Fe(III) from Mn(II). It is also worth noting here that the pH of the receiving phase became more acidic (from pH 5 to pH 1.5–2) as the time was elapsed. This result can be attributed to the co-transport of HCl from the feed to the receiving phase (see Section 3.2).

#### 3.2.2. Influence of the Stirring Speed Applied on the Receiving Phase on Iron(III) Transport

The influence of the stirring speed on iron(III) transport, applied to the receiving phase, was also investigated using the same experimental conditions showed in Table 1, albeit using a stirring speed in the feed phase of 1250 min^−1^ and varying the stirring speed in the receiving phase from 500 to 1000 min^−1^ The results showed that this variation had no influence on iron(III) transport or in the recovery of the element in this phase. This behavior was attributable to that if the stirrer in the receiving phase half-cell was very close to the membrane support, the thickness of the boundary layer was considered to be minimized, and the resistance in this side can be neglected [21].

#### 3.2.3. Influence of the HCl Concentration in the Feed Phase on Iron(III) Transport

The influence of the variation of the acid concentration in the feed phase on iron(III) transport was investigated using the same experimental conditions showed in Table 1, but with acid concentrations ranging from 1 to 8 M. In these series of experiments, the stirring speeds applied on the feed and receiving phases were of 1250 and 500 min^−1^, respectively. The results from these series of experiments were summarized in Table 2. It can be seen that iron(III) transport increased when the HCl concentration increased in the 1–4 M range, and then decreased at HCl concentrations of 6 and 8 M. This decrease can be explained due to the competitive transport of HCl across the supported liquid membrane impregnated with Cyanex 923 dissolved in Solvesso 100, it was shown [22] that Cyanex 923 extracted mineral acids from aqueous solutions. In the case of manganese(II), negligible transport of this element was detected along the experiments.

#### 3.2.4. Influence of Cyanex 923 Concentration on Iron(III) Transport

Previously, it was detected that the solid support impregnated only with Solvesso 100 did not transport neither iron(III) or manganese(II). Thus, the presence of the extractant or carrier was essential to achieve metal transport from the feed to the receiving solution. This influence was investigated using a feed phase of 0.01 g/L Fe(III) and 0.17 g/L Mn(II) in 4 M HCl medium, whereas the receiving phase was of distilled water; in these series of experiments, the extractant concentration was varied from 1.3 to 40% *v/v* (0.03–1 M) dissolved in Solvesso 100. The results of these experiments are summarized in Table 3.

Results showed in the above Table 3 indicated that there was an increase of iron(III) permeation with the increase from 1.3 to 10% *v/v* Cyanex 923 concentration in the carrier solution, and then a decrease in iron(III) transport resulted when higher Cyanex 923 concentrations (20–40% *v*/*v*) were used. This decrease, at a first instance, can be attributed to the increase of the carrier phase viscosity, which resulted in lower values of the overall mass transfer coefficient. In the case of manganese(II), negligible transport was observed, even using the most concentrated Cyanex 923 solution. At 10% *v/v* Cyanex 923 in Solvesso 100 concentration, a maximum in iron(III) transport was attained, this maximum or limiting mass transfer coefficient (K_lim_) was explained by the assumption that in this experimental condition, the diffusion in the membrane (Δ_m_) was negligible in comparison with feed diffusion (Δ_f_), and the transport process was entirely controlled by the diffusion in the film of the feed solution. Thus:(7)$$K_{\lim}=\frac{1}{\Delta_{f}}=\frac{D_{f}}{d_{f}}$$

In the above equation, D_f_ = 10^−5^ cm^2^/s represented the average diffusion coefficient in the aqueous feed phase, and d_f_ to the thickness of the feed phase boundary layer. Accordingly, in the present system, d_f_ was calculated as 2.9 × 10^−3^ cm. This value represents the minimum thickness of the boundary layer under the present experimental conditions.

#### 3.2.5. Influence of Manganese(II) Concentration in the Feed Phase on Iron(III) Transport

The possible variation of manganese(II) concentration in the feed phase on iron(III) transport was also investigated. The carrier phase was of 10% *v/v* Cyanex 923 in Solvesso 100 impreganting Durapore GVHP4700 support, whereas the receiving solution was water. The aqueous feed phase contained 0.01 g/L Fe(III) in 4 M HCl medium and varying Mn(II) concentrations from nil to 0.17 g/L. The results indicated that the presence of Mn(II) in the solution had not detrimental effect on iron(III) transport (K_Fe_ = 3.2 ± 0.2 × 10^−3^ cm/s), thus, crowding or population effect [23], due to the presence of Mn(II) in the solution, was negligible on iron(III) permeation under these experimental conditions.

#### 3.2.6. Estimation of Diffusional Parameters and Evaluation of Mass Transfer Resistances

The permeation rate of iron(III) was estimated by the rates associated to the diffusion of Fe(III)-bearing species through the feed phase diffusion layer and the diffusion of the Fe(III)-Cyanex 923 species through the membrane. Fick’s first diffusion law was applied to the diffusion layer in the feed and the membrane sides [24]. The fluxes in each of the above phases can be represented by:(8)$$J_{f}=\Delta_{f}^{-1}\left({\left[\mathrm{Fe}\right]}_{f,0}-{\left[\mathrm{Fe}\right]}_{f,i}\right)$$
(9)$$J_{m}=\Delta_{m}^{-1}\left({\left[H\mathrm{Fe}Cl_{4}L_{4}\right]}_{f,i}-{\left[H\mathrm{Fe}Cl_{4}L_{4}\right]}_{r,i}\right)$$
where [Fe]_f,0_ was the initial iron concentration in the feed phase, [Fe]_f,i_ was the iron concentration at the feed phase/membrane interface, and [HFeCl_4_L_4_^−^]_f,i_ and [HFeCl_4_L_4_]_r,I_ were the concentrations of the species at the feed phase/membrane interface and the membrane/receiving phase interface, respectively.

Usually, the concentration of the metal-organic ligand species (HFeCl_4_L_4_ in the present investigation) in the membrane phase at the receiving phase side was negligible when compared with the corresponding one at the feed phase side. Thus, Equation (9) was rewritten as:(10)$$J_{m}=\Delta_{m}^{-1}{\left[HFeCl_{4}L_{4}\right]}_{f,i}$$

It can be considered that the chemical reaction (Equation (3)) was fast if compared to the diffusion rate, thus, local equilibrium at the feed phase-membrane interface was attained and concentrations in this interface was related by Equation (4). At steady state, J = J_f_ = J_m_, being J expressed as:(11)$$J=K_{Fe}{\left[Fe\right]}_{f,0}$$

By combination of Equations (4), (8), (10) and (11), the following expression was derived:(12)$$K_{Fe}=\frac{K_{ext}{\left[L\right]}_{m}^{4}{\left[H^{+}\right]}_{f}}{\Delta_{m}+\Delta_{f}\left(K_{ext}{\left[L\right]}_{m}^{4}{\left[H^{+}\right]}_{f}\right)}$$
where Δ_f_ and Δ_m_ were the transport resistances due to diffusion by the feed phase boundary layer and the membrane, respectively. 

This last equation combined in one expression the equilibrium and diffusional parameters involved in the iron(III) transport though a flat-sheet supported liquid membrane containing Cyanex 923 in Solvesso 100 as carrier phase.

The resistances to the mass transfer were evaluated by the use of Equation (12), from this equation the next expression can be derived:(13)$$\frac{1}{K_{Fe}}=\Delta_{f}+\frac{\Delta_{m}}{K_{ext}{\left[L\right]}_{m}^{4}{\left[H^{+}\right]}_{f}}$$

A plot of 1/K_Fe_ versus 1/K_ext_[L]_m_^4^[H^+^]_f_, for various extractant concentrations in Solvesso 100 and an aqueous feed solution of 4 M HCl, allowed to determine Δ_m_ (slope) and Δ_f_ (ordinate). From the above plot (r^2^ = 0.978), it was determined that Δ_m_ = 145 s/cm and Δ_f_ = 361 s/cm. These values are the transport resistances due to diffusion through the membrane pores and the feed boundary layer, respectively.

The diffusion coefficient in the membrane phase is defined as
(14)$$D_{m}=\frac{d_{m}}{\Delta_{m}}$$
was calculated as 8.6 × 10^−5^ cm^2^/s. In the above equation, d_m_ represented the membrane thickness. The diffusion coefficient of the phosphine oxide-iron(III) species in the bulk organic phase [25]:(15)$$D_{b,m}=D_{m}\frac{\tau^{2}}{\varepsilon }$$
was estimated as 3.2 × 10^−4^ cm^2^/s. This value was greater than the value of the diffusion coefficient in the membrane phase (Equation (14)), this was attributable to the diffusional resistance caused by the membrane thickness, separating the feed and the receiving phases, In Equation (15), τ is the membrane tortuosity (1.67) and ε is the support porosity (75).

Assuming that the extractant concentration in the membrane phase was constant, an apparent diffusion coefficient for iron (III)-Cyanex 923 dissolved in Solvesso 100 species was estimated as:(16)$$D_{m}^{a}=\frac{Jd_{m}}{{\left[L\right]}_{m}}$$
where J was calculated accordingly to Equation (11) and an initial iron(III) concentration in the feed phase of 0.01 g/L; considering an extractant concentration of 10% *v/v* Cyanex 923 in Solvesso 100, this apparent coefficient van be calculated as 3 × 10^−8^ cm^2^/s.

Since the overall mass transfer resistance was the result of the sum of the different resistances participating in the permeation process, Equation (13) can be rewritten as:(17)$$R=R_{f}+R_{m}$$
where R_f_ and R_m_ were the resistances in the feed and the membrane phases, respectively. Table 4 showed the contribution of these resistances to the iron(III) transport.

Furthermore, the fractional resistance of each step (%R_f_^0^ and %R_m_^0^) to the transport process was calculated under various experimental conditions. As it can be seen from the results presented in Section 4, in almost all of the conditions, diffusion by the aqueous feed boundary layer was the controlling step for the iron(III) transport process.

## 4. Conclusions

The use of Cyanex 923, dissolved in Solvesso 100, coupled to a solid supported liquid membrane (flat-sheet mode) technology, allowed the selective separation of Fe(III) over Mn(II) from solutions derived in the treatment of spent alkaline batteries. A kinetic model for Fe(III) transport is developed and finding that Fe(III) transport is mainly controlled by diffusion of protons and FeCl_4_^-^ species across the feed aqueous layer. This contributes, under certain experimental conditions, to the diffusion by the membrane of the neutral complex formed by the HFeCl_4_ species and Cyanex 923.

## Figures and Tables

**Figure 1 membranes-11-00991-f001:**
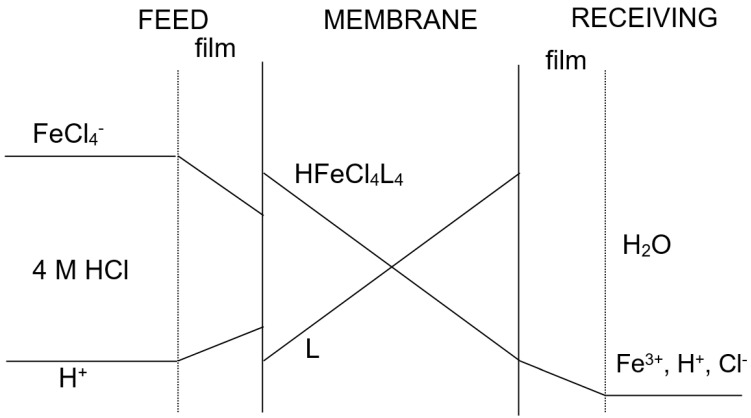
Concentration profile of the species through the supported liquid membrane.

**Table 1 membranes-11-00991-t001:** Influence of the stirring speed on transport of Fe(III) as a function of the overall mass transfer coefficient (K_Fe_).

Stirring Speed, min^−1^	K_Fe_ 10^3^, cm/s	% Fe Recovery ^a^
500	0.7	99
1000	1.1	99
1250	1.4	99
1500	1.5	99

Feed phase: 0.01 g/L Fe(III) and 0.17 g/L Mn(II) in 2 M HCl. Feed phase stirring speed: variable. Membrane phase: 10% *v/v* Cyanex 923 in Solvesso 100 supported on GVHP4700. Receiving phase: distilled water. Receiving phase stirring speed: 500 min^−1^. Temperature: 20 °C. ^a^ Recovery in the receiving phase after 3 h.

**Table 2 membranes-11-00991-t002:** Influence of HCl concentration on iron(III) transport.

[HCl], M	K_Fe_·10^−3^, cm/s	% Fe Recovery ^a^
1	0.08	99
1.5	0.56	99
2	1.4	99
4	3.4	72
6	3.1	68
8	2.3	65

^a^ Recovery in the receiving phase after 3 h.

**Table 3 membranes-11-00991-t003:** Influence of Cyanex 923 concentration in the carrier phase on iron(III) transport.

[Cyanex 923], % *v*/*v*	K_Fe_·10^−3^, cm/s	% Fe Recovery ^a^
1.3	1.0	90
2.5	2.3	99
5	2.5	98
10	3.4	72
20	2.9	71
30	2.3	72
40	2.1	72

^a^ Recovery in the receiving phase after 3 h.

**Table 4 membranes-11-00991-t004:** Contribution of mass transfer resistances to iron(III) transport.

Experimental Condition	^a^ R, s/cm	^b^ R_f_, s/cm	%R_f_^0^	%R_m_^0^
1–10% *v/v* Cyanex 923	1000–294	361	36–100	64–0
10–40% *v/v* Cyanex 923	294–476	361	100–76	0–24
1–4 M HCl	12,195–294	361	3–100	97–0
4–8 M HCl	294–435	361	100–83	0–17
0–0.17 g/L Mn(II)	294	361	100	0

^a^ Experimental values. ^b^ Model value.

## Data Availability

All the data supporting the findings of this study are available within the article.

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
