# Peer review of "Separation Iron(III)-Manganese(II) via Supported Liquid Membrane Technology in the Treatment of Spent Alkaline Batteries"

_membranes, 2021, doi:10.3390/membranes11120991_

Round 1
Reviewer 1 Report
In this manuscript, the authors developed a flat-sheet supported liquid membrane to successively separate iron(III) from the mixed solutions of Fe(III)-Mn(II)-HCl, anticipated to use in the spent alkaline batteries. A commercially available carrier phase of Cyanex 923 was used in the SLM system by dissolving in Solvesso 100 diluent (which is also commercially available) and performed at hydrodynamic conditions in which the feed phase was measured concentration of manganese and HCl and carrier concentration in the membrane phase to analyze the selective transport of iron(III) through the SLM. Although the authors performed relevant characteristic studies and discussed key parameters toward separation of iron(III), the authors need to address more detail the concept and relevance of the present study in membrane-based separation technology.
Comments:
- The introduction seems very simple and does not have much information regarding the relevance of the current study. Most of the paragraphs in the introduction part are just three or four lines!!!
- In methods 2.2.1, the authors did not cite the reference, please explain in detail with the reference.
- In section 3.2, the authors need to emphasize more on the stoichiometric factor n and need to clarify what is the benefit as it was obtained to a value of 4
- The authors need to rewrite section 3.3.1 to better understand what was being discussed in this section. Also, correct typos and grammar mistakes.
- The authors mentioned that it can be seen that the overall mass transfer value increased with the increase of the stirring speed up to 1250 min-1. However, the Fe recovery found the same in all stirring speeds, please explain how the acidic HCl drastically increased although Fe carrier Cyanex 923 amount is still in a good amount.
- The conclusion does not convey the knowledge and application perspectives. Please emphasize more and conclude nicely!
- Seriously, the authors must check the manuscript carefully for grammatical errors.
Author Response
In this manuscript, the authors developed a flat-sheet supported liquid membrane to successively separate iron(III) from the mixed solutions of Fe(III)-Mn(II)-HCl, anticipated to use in the spent alkaline batteries. A commercially available carrier phase of Cyanex 923 was used in the SLM system by dissolving in Solvesso 100 diluent (which is also commercially available) and performed at hydrodynamic conditions in which the feed phase was measured concentration of manganese and HCl and carrier concentration in the membrane phase to analyze the selective transport of iron(III) through the SLM. Although the authors performed relevant characteristic studies and discussed key parameters toward separation of iron(III), the authors need to address more detail the concept and relevance of the present study in membrane-based separation technology.
Thank you for your comments about the work. Find below our response to them.
Comments:
The introduction seems very simple and does not have much information regarding the relevance of the current study. Most of the paragraphs in the introduction part are just three or four lines!!!
What is wrong with paragraphs of three-four lines?, this is our style of writing here. Please note that along the Introduction, we concisely indicated the main point in which our investigation is based, and also very recent references on the use of supported liquid membranes in the removal of metals from solutions.
In methods 2.2.1, the authors did not cite the reference, please explain in detail with the reference.
Though you do not mention where the reference must be, we assumed that is in relation with the sentence: Previous experiments showed that equilibrium was reached within 5 min of contact between both phases. As you and readers must understand it means that these experiments were carried out in the context of the present investigation, but the overall data, considered as superfluous for us, were not included in the manuscript, just a sentence is needed here.
In section 3.2, the authors need to emphasize more on the stoichiometric factor n and need to clarify what is the benefit as it was obtained to a value of 4
This factor n was derived from the experimentation and the treatment of the data by the computer program, thus, it is valid under the present experimental conditions; the change of these conditions can affect to the metal-extractant stoichiometry.
There is not any benefit with this factor of 4, it is just a number.
The authors need to rewrite section 3.3.1 to better understand what was being discussed in this section. Also, correct typos and grammar mistakes.
An English speaking person had read the section and think that it is correct, though not easy to understand to people, as he is, with very limited knowledge on supported liquid membranes science.
The authors mentioned that it can be seen that the overall mass transfer value increased with the increase of the stirring speed up to 1250 min-1. However, the Fe recovery found the same in all stirring speeds, please explain how the acidic HCl drastically increased although Fe carrier Cyanex 923 amount is still in a good amount.
What dou you want to say with acidic HCl? It has no sense.
Moreover, what do you wan to say with acidic HCl drastically increased although Fe carrier Cyanex 923 amount is still in a good amount? Also, this sentence has not sense
Moreover, we do not understand the relationship between the first sentence: The authors mentioned…., and the second sentence: However, the Fe recovery…..
The conclusion does not convey the knowledge and application perspectives. Please emphasize more and conclude nicely!
We think that as it is written, the Conclusion section concisely addressed the relevant results derived from the experimental work. Not superfluous or repetition of data here and in relation to what is given along the manuscript.
Seriously, the authors must check the manuscript carefully for grammatical errors.
Thank you
Reviewer 2 Report
Authors reported separation Fe(III)-Mn(II) via supported liquid membrane technology in the treatment of spent alkaline batteries. Work is interesting. However, it is acceptable after the incorporation of the following raised concerns.
- Symbol X should be replaced with a Multiply symbol in Table 1 and throughout the manuscript.
- Section 3.1 there is an iron or irond in the heading?
- What is the role of Mn (II) in this study?
- The authors added limited discussion on results. Results should be elaborated more with the evidences.
- In the title and abstract, full names should be given along with symbols
- In the title, authors mentioned batteries but did not mention in the abstract and conclusion and even in the discussion of the results
- The introduction is too short
- Reagents and solutions have no details from which company and country have been bought
- The membrane diffusional resistance (Δm) and feed diffusional resistance (Δf) were calculated, and their values are 145 s/cm and 361 s/cm. What is the significance of these values? Compare with the recently reported literature and insert Table.
Author Response
Authors reported separation Fe(III)-Mn(II) via supported liquid membrane technology in the treatment of spent alkaline batteries. Work is interesting. However, it is acceptable after the incorporation of the following raised concerns.
Thank you for your comments, please find below our respective responses to them
Symbol X should be replaced with a Multiply symbol in Table 1 and throughout the manuscript.
Done
Section 3.1 there is an iron or irond in the heading?
Amended
What is the role of Mn (II) in this study?
As we mentioned in the Introduction Section, the acidic leaching step of the material leads to a solution containing Fe(III) and Mn(II), since the goal of the operation is to separate both metals, this is why manganese(II) appeared in the aqueous solutions used throughout the work
The authors added limited discussion on results. Results should be elaborated more with the evidences.
Here we does not agree with you, we think that the discussion is properly done in each section
In the title and abstract, full names should be given along with symbols
Done
In the title, authors mentioned batteries but did not mention in the abstract and conclusion and even in the discussion of the results
Added in the abstract and at the end of the Introduction section
The introduction is too short
Here we do not agree with you, we think that it is adequate because it presented concisely the problem (the treatment of residues as spent alkaline batteries are), how we processed them, objective of the present investigation, etc.
Reagents and solutions have no details from which company and country have been bought
Added
The membrane diffusional resistance (Δm) and feed diffusional resistance (Δf) were calculated, and their values are 145 s/cm and 361 s/cm. What is the significance of these values? Compare with the recently reported literature and insert Table.
Evidently these diffusional resistances indicated the resistance of each step to diffuse the solute (in the present case HFeCl4-extractant (membrane) or H+ and FeCl4- (feed solution) species) across the membrane or the aqueous feed boundary layer, respectively. Since the literature used different carriers and/or solutes, and very different experimental conditions, the comparison between the different resistances values, obtained under these different conditions, added little to the overall knowledge about supported liquid membrane systems.
Round 2
Reviewer 1 Report
Please find the attached file!

Author Response
Thank you for your comments about the work. Find below our response to them.
The authors’ responses to the comments and suggestions are not adequate. The authors should improve their manuscript from reviewers’ valuable comments rather than defending/arguing with irrelevant disputes.
Please note that reviewer 2 agreed with our comments and/or changes and accepted the revised version 1 of the manuscript.
Comments:
The introduction seems very simple and does not have much information regarding the relevance of the current study. Most of the paragraphs in the introduction part are just three or four lines!!!
What is wrong with paragraphs of three-four lines?, this is our style of writing here.
You can follow your style but need to be strictly abided by the journal’s format and instructions to prepare manuscripts. The authors must take reviewers’ suggestions seriously to improve their manuscripts to convey knowledge to the MDPI journal readers. The authors must read “Instructions for Authors” and prepare how to write a scientific journal.
Sorry but we do not agree with you, after publishing more than 300 articles in SCI Journals (including 21articles in MDPI Journals) you can not argue the above.
https://www.mdpi.com/journal/membranes/instructions#preparation
- Introduction: The introduction should briefly place the study in a broad context and highlight why it is important. It should define the purpose of the work and its significance, including specific hypotheses being tested. The current state of the research field should be reviewed carefully and key publications cited. Please highlight controversial and diverging hypotheses when necessary. Finally, briefly mention the main aim of the work and highlight the main conclusions. Keep the introduction comprehensible to scientists working outside the topic of the paper.
Unfortunately, the present manuscript does not meet the basic requirements of plotting a nice introduction. Work on it nicely!!!
The same comments as above
Please note that along the Introduction, we concisely indicated the main point in which our investigation is based, and also very recent references on the use of supported liquid membranes in the removal of metals from solutions.
In methods 2.2.1, the authors did not cite the reference, please explain in detail with the reference.
Though you do not mention where the reference must be, we assumed that is in relation with the sentence: Previous experiments showed that equilibrium was reached within 5 min of contact between both phases. As you and readers must understand it means that these experiments were carried out in the context of the present investigation, but the overall data, considered as superfluous for us, were not included in the manuscript, just a sentence is needed here.
This sentence is absurd, the sentence “previous experiments showed…” misinform your experimental analyses. Change the sentence something like “After the careful optimization of experimental setups, it was found that equilibrium reaches within 5 minutes of contact between both the phases”.
We use the sentence kindly provided by yours.
In section 3.2, the authors need to emphasize more on the stoichiometric factor n and need to clarify what is the benefit as it was obtained to a value of 4
This factor n was derived from the experimentation and the treatment of the data by the computer program, thus, it is valid under the present experimental conditions; the change of these conditions can affect to the metal-extractant stoichiometry.
There is not any benefit with this factor of 4, it is just a number.
I was trying to say that are there any benchmarks to show a trade-off relation between stoichiometric factor (n) and experimental conditions. If that is the case, explain how the obtained n value (i.e., 4) affects the metal extraction.
Again, note that it is a stoichiometric factor, that indicated the molar relationship between the extractant and iron species in the extracted complex, under the present experimental conditions. We considered irrelevant to add the above to the manuscript, it did not add nothing
The authors need to rewrite section 3.3.1 to better understand what was being discussed in this section. Also, correct typos and grammar mistakes.
An English speaking person had read the section and think that it is correct, though not easy to understand to people, as he is, with very limited knowledge on supported liquid membranes science.
The authors must acknowledge that respective English native for his/her great help to improve this manuscript.
Again, read the instructions.
We include the namei in the revised manuscript
- Discussion: Authors should discuss the results and how they can be interpreted in perspective of previous studies and of the working hypotheses. The findings and their implications should be discussed in the broadest context possible and limitations of the work highlighted. Future research directions may also be mentioned. This section may be combined with Results.
The conclusion does not convey the knowledge and application perspectives. Please emphasize more and conclude nicely!
We think that as it is written, the Conclusion section concisely addressed the relevant results derived from the experimental work. Not superfluous or repetition of data here and in relation to what is given along the manuscript.
Seriously, the authors must check the manuscript carefully for grammatical errors.
Thank you
I have run a plagiarism checker and found that there are not many changes in the present manuscript from a recent preprint version. Of course, it is not a self-plagiarism to develop a full-length article out of preprints, but it is highly encouraged to make substantial changes from the preprint version.
In our opinion, the manuscript version must not be different or very different of the preprinted version, thus, the mention of plagiarism or even self-plagiarism is out of context and seemend to be a little offensive against us. Accordingly, we maintained the Conclusions as such.

Reviewer 2 Report
The authors have provided the suggested changes. The manuscript is suitable for publication.
Author Response
Thank you for your positive comments about our revised version and the acceptance of the manuscript to be published. Though you mentioned in your review that you like to sign your review, the Editorial Office does not include your name in the reviewer reply.